# Effect of Therapeutic Gardening Program in Urban Gardens on the Mental Health of Children and Their Caregivers with Atopic Dermatitis

**DOI:** 10.3390/healthcare12090919

**Published:** 2024-04-29

**Authors:** Hyunjin Baik, Sueran Choi, Miae An, Hyeyoung Jin, Insoon Kang, Wonsuck Yoon, Young Yoo

**Affiliations:** 1Korea Research and Institute for People & Environment, Seoul 05737, Republic of Korea; jina100@kripe.co.kr (H.B.); info@kripe.co.kr (S.C.); admin@kripe.co.kr (M.A.); 2Korea National Arboretum, Pocheon 11186, Republic of Korea; jinhye0@korea.kr; 3Allergy Immunology Center, Korea University, Seoul 02841, Republic of Korea; aesim99@kumc.or.kr (I.K.); biokorea@korea.ac.kr (W.Y.); 4Department of Pediatrics, Korea University Anam Hospital, Seoul 02841, Republic of Korea

**Keywords:** garden therapy, healing garden, psychosocial intervention, parenting efficacy, nature-based intervention

## Abstract

This study was conducted to identify the physical and psychological difficulties of children with atopic dermatitis and their caregivers, and to investigate the effects of participation in a novel therapeutic gardening program based on psychological intervention on their physical and mental health. The program, consisting of 15 sessions, was conducted for approximately 4 months in urban gardens in Seoul and involved 30 children with atopic dermatitis and their caregivers. Additionally, a control group of 30 non-participating caregivers was recruited for comparative analysis. The psychological and emotional changes in caregivers were assessed using six self-report scales (depression, anxiety, stress, vitality, life satisfaction, parenting efficacy) before and after participation in the program. Additionally, the depression index (CDI) and atopic dermatitis index (SCORAD, TEWL) were measured for the children with atopic dermatitis. The research results indicate that the therapeutic gardening program utilizing psychological intervention had a positive impact on the physiological and psychological health of participants. These results are significant as they demonstrate the clinical application of the professionally developed therapeutic gardening program through active intervention and operation. This study suggests that this program can serve as an effective intervention in improving the mental health of both children with atopic dermatitis and their caregivers.

## 1. Introduction

Atopic dermatitis (AD) is a common, chronic, and recurrent disorder in modern society, significantly influencing not only the quality of life of the affected child but also that of the entire family [1,2,3]. According to an announcement by the National Statistical Office of Korea in 2021, 24.5%, 21%, and 23.9% of elementary, middle, and high school students, respectively, were diagnosed with AD, and the proportion of severely ill patients is gradually increasing (Statistics Korea, 2021). Due to the challenges in treating and managing AD, which is not easily cured, parents often experience negative psychological states and sleep disorders along with their children [4,5,6]. This not only limits daily life but also intensifies the caregiving burden, giving rise to fatigue, despair, guilt, depression, and other challenges in caregivers [7]. Related research conducted on children with AD and their caregivers has confirmed that mothers of children with AD suffered from higher parenting stress and depression than mothers of normal children [5,8]. Caregivers with a tendency toward melancholy do not respond quickly to their children’s needs and lack the ability to nurture them, leading to a deterioration in their relationships with their children [9]. Conversely, caregivers with low stress and depression levels have high parenting efficacy and help their children’s emotional stability through active interaction and empathy [10,11,12]. Mothers’ parenting stress, efficacy, and interactions with their children also correlate [13]. This compromised psychological state not only diminishes parenting efficacy but also contributes to increased depression and stress levels in the affected children, hindering the improvement of atopic dermatitis symptoms. Therefore, participating in a TGP in an urban garden setting, where natural interaction is possible and the recovery effect is high, is expected to help transform negative psychological states, such as depression, anxiety, and stress, into positive ones. Through this, such programs can enhance parenting efficacy and vitality, ultimately improving overall quality of life.

Environmental and psychological factors can also worsen symptoms in children with AD [14,15,16]. Stress in daily life can weaken the immune system [17] and negatively impact AD symptoms [18,19,20]. Furthermore, the caregiver’s parenting stress and negative psychological state can also act as environmental factors impacting interaction with the child, potentially exacerbating the symptoms of atopic dermatitis in the child [16,21]. For these reasons, psychological factors have gained attention as significant regulatory factors in AD [19,22,23,24,25]. Therefore, beyond treating the skin lesions to improve symptoms in children with AD, various management strategies are needed for stress management and mental health enhancement. This includes the introduction of diverse programs to enhance interaction with caregivers and improve parenting efficacy, addressing issues such as depression and anxiety. Therefore, we planned activities in a garden setting that allows for psychosocial interventions as a program to alleviate psychological distress and stress in both children with AD and their caregivers. Gardens provide an optimal recovery environment for health promotion [26,27,28,29] and well-being, and activities in gardens have a positive effect on improving physical health [30,31] and psychological stability [32,33,34,35,36,37,38,39] through active interaction with nature. Gardens in the city center allow for easier contact with nature in everyday life and have the advantage of being able to engage people in gardening activities in a relatively small space [40]. Moreover, from a medical standpoint, participation in gardening activities has been found to improve physical and mental well-being [7,23,41,42,43], including preventing depression [44] and reducing levels of physiological stress indicators [45,46] such as cortisol, heart rate, and blood pressure. 

In this study, we aimed to enhance the mental health of children with AD and their caregivers (reducing depression, anxiety, and stress, increasing life satisfaction, vitality, and parenting efficacy). To achieve this, we developed a total of 15 sessions of a Therapeutic Gardening Program (TGP) and implemented it in an urban garden. The TGP was designed to facilitate active interaction and communication between children and caregivers through garden activities, with a focus on promoting psychological stability and stress relief and increasing the parenting efficacy of caregivers. 

The hypotheses are as follows: Firstly, the planned TGP is expected to positively change the negative psychological state (depression) of children with AD and their caregivers (depression, anxiety, and stress). Secondly, participation in the TGP is anticipated to induce positive changes in caregivers’ vitality and life satisfaction and enhance parenting efficacy. Thirdly, it is expected that there is a correlation between caregivers’ mental health variables and parenting self-efficacy. Fourthly, the increase in parenting efficacy and the positive psychological state of caregivers is expected to be associated with a decrease in the child’s depression. Finally, the TGP, which was designed to promote interaction and communication among participants, is expected to positively influence mental health variables and increase parenting self-efficacy, effectively enhancing the psychological health and well-being of participants.

## 2. Materials and Methods

### 2.1. Participants

This study received approval from the Public Institutional Review Board (P01-202206-01-011) designated by the Ministry of Health and Welfare. A total of 60 participants were recruited from caregivers of children diagnosed with AD, who were experiencing psychological and emotional difficulties. The inclusion criteria comprised caregivers without allergies to sunlight, pollen, and similar allergens, who could participate in outdoor and gardening activities without difficulty for an extended period. Additionally, caregivers who were taking antidepressant medications or undergoing other forms of psychological therapy during the intervention period were excluded. Caregivers recruited for the study were randomly assigned to either the experimental group (30 participants) or the control group (30 participants). The children whose caregivers were assigned to the experimental group (30 children) were automatically enrolled in the program. Informed consent from parents/caregivers was obtained, based on the information provided, as the study involved participants under the age of 18. The Consort Flow Diagram in Figure 1 shows more details about the allocation to conditions. The number of study subjects was calculated using the G*Power 3.1.9.7 (Heinrich Heine Universität Düsseldorf, Düsseldorf, Germany) program (considering a dropout rate of 10%, the effective size of 0.5, a power (β) of 0.8, and a significance level (α) 0.05). To recruit participants, posters containing the research purpose, TGP contents, and evaluation methods were posted on online bulletin boards with the cooperation of the Korea University Anam Hospital and the Gwangjin-gu Public Health Center. Written informed consent was obtained from all the parents and caregivers of all participants.

### 2.2. Procedure

This study was conducted to investigate the effect of participation in a TGP on the psychosocial health and improvement of symptoms of children with AD and their caregivers. The TGP was conducted from July to October 2022 for approximately 4 months (15 sessions in total) at Ttukseom Hangang Park. The activities consisted of all the activities included in gardening, from garden creation to garden maintenance, management, and utilization. Furthermore, the program was designed to foster active interaction and communication between children and caregivers, aiming to achieve psychological stability through the TGP. A pre-test for depression was conducted before the program was implemented, and a quasi-experimental design was used to test the effects. Participants’ psychosocial changes were measured before the start of the program (first session) and at the end of the program (last session) (Appendix A).

Each session was structured into stages, following the sequence of “Preparation-Warm-up-Rest-Main activity-Group feedback-Closure”, with an average duration of approximately 3 h (Appendix A). The Preparation stage served as an initial step for program execution, involving attendance checks and confirming participants’ conditions to tailor the day’s activities accordingly. The Introduction stage was designed to alleviate tension and facilitate a comfortable start to the activities, including sharing missions and activity content from previous sessions. Additionally, a workbook was utilized to organize thoughts, emotions, and missions related to the theme of the session. After a brief rest, participants moved to the garden, engaging in light walking and relaxation exercises. In the main activity stage, gardening activities related to the introduced theme were conducted which were designed to immerse participants and induce psychological relaxation. Following the activity, a group feedback session was held to discuss results and share emotions. Participants were encouraged to support and motivate each other during the discussion.

### 2.3. Therapeutic Gardening Program Design

#### 2.3.1. Program Site

The TGP was conducted at the Healing Garden located in Hangang Park in Seoul. The Healing Garden (206 m^2^) is situated adjacent to various spaces, including the Rose Garden (6040 m^2^, over 40 varieties of roses), the Nature Learning Area (23,100 m^2^, over 210 varieties of plants), and the Cypress Forest (5000 m^2^, over 600 coniferous trees), enabling diverse program activities that make use of the nearby natural environment. Additionally, spaces for rest during the garden activities were provided, including an indoor activity space within the information center, pergolas, pavilions, and benches (Figure 2).

#### 2.3.2. Program Goal-Setting

Based on previous research, a preliminary survey was conducted to identify the psychosocial difficulties of children with AD and their caregivers, and goals were set accordingly. The objective of the TGP is to reduce stress and increase parenting efficacy, ultimately contributing to the improvement of the family’s quality of life, as predicted in Figure 3. The program was designed as an intervention strategy to enhance the interaction, understanding, and emotional support between the children and their caregivers through the TGP. Additionally, by examining changes in atopic symptoms in children with AD, we confirmed the possibility of improving AD symptoms through improvements in mental health.

#### 2.3.3. Program Composition

The TGP for children with AD and their caregivers was meticulously designed to encompass horticulture and gardening activities, physical activity, tasks fostering a sense of accomplishment, and active interaction with the child. By employing emotional support and empathetic interaction as mediation strategies, our goal was to enhance parenting efficacy and mitigate depression, anxiety, and stress through gardening activities, and to stimulate interaction with children by fostering self-understanding and communication (Figure 4). The program was designed with a step-by-step approach to enable participants to highlight their strengths and abilities and find their own values, leading to positive changes.

Prior to the program planning, we categorized and detailed the garden activities, as depicted in Figure 5. Our objective was to design various activities within the garden for parents and children, ultimately aiming to facilitate their transition to a positive psychological state through garden activities.

We incorporated Teti and Gelfand’s (1991) [47] parenting efficacy factors, such as helping children understand parents’ expectations, recognizing preferences, and facilitating interaction, into the garden activities to improve overall parenting efficacy and competence.

During each session, we identified current factors of depression and stress, provided opportunities for interaction, and presented participants and their children with the mission of finding factors that could help them overcome negative psychological states. These missions included activities using garden plants and herbs, discovering healing plants in the city, and creating a garden that attracts butterflies.

The TGP established goals for each session based on the activity sequence and content outlined in the plan (Appendix A), progressing accordingly. The TGP operation involved dividing participants into small groups of 3–4 individuals, with a dedicated gardener assigned to each group. This approach allowed for (1) the alleviation of congestion through small-scale gardening activities, (2) enabling participant observation and quick feedback, (3) aiding in the analysis of participant behavior changes, and (4) facilitating the smooth observation and recording of participants during each session. To ensure consistent implementation, training sessions for gardeners and assistant staff were conducted prior to the commencement of the TGP, allowing for the sharing of activity goals and operational plans. The feedback from each session was used to concretize and incorporate improvement points in the subsequent sessions.

#### 2.3.4. A Phased Plan for the TGP

The TGP was planned in stages according to the healing goals. The first step was “Garden for Start”, which was planned to create intimacy and belonging. This is the basic stage of the program in which participants plan the planting plan and create a garden. The purpose of this stage is to provide a direct experience with natural elements, impart a basic understanding of the garden and plants they will be working with, and foster familiarity and understanding among participants. During this process, we planned to actively communicate with nature through direct contact with soil and frequent contact with plants. Planting a family tree with the child and creating a family garden space helps to promote interaction and provides a shared mission to facilitate communication.

The second stage was “Garden for Participation”, which aimed for active participation in gardening. This stage was planned to discover self-value and realize self-understanding through plants. While creating a garden in earnest, one can discover their strengths and values through the telling of various stories about plants and communicate and understand garden management with children.

The third step was “Garden for Experience”, which aimed to improve inner confidence and bring about change through shared gardening experiences. This stage was planned to create various experiences in the garden with the family. It was designed to remove negative emotions and seek change through activities together in the family garden space. The activities included finding and breeding plants to increase time with children and provide opportunities for natural conversation. 

The final step was “Garden for Restart”, which aimed to encourage participants to continue gardening at home after the program had ended. Seed collection, winter preparation, and plant transplantation were carried out to strengthen the benefits discovered through the program, enhance communication with the children, and continue gardening activities at home.

### 2.4. Psychosocial Effect Measures

#### 2.4.1. Mental Health Screening Tool for Depressive and Anxiety Disorders

Depression and anxiety were evaluated using the ‘Korea Mental Health Screening Tool for Depression (MHS:D) and Anxiety (MHS:A)’, respectively. Both are self-reported assessment tools asking how much an individual has experienced symptoms related to major depression and anxiety disorder in the past 2 weeks. MHS:D and MHS:A consist of a 5-point Likert scale (0 = never; 4 = very much).

The MHS:D questionnaire (12 items) includes questions related to depression, such as depressed mood, decreased interest, fatigue, worthlessness, hopelessness, appetite or weight changes, and sleep changes. A total score of 0–8 is considered minimal, 8–12 as mild, 12–20 as moderate, and >20 as severe [48]. The reliability coefficient of Cronbach’s α of the evaluation tool is 0.94, and the reliability coefficient in this study was 0.83.

The MHS:A questionnaire (11 items) includes generalized anxiety disorder diagnostic domains, such as excessive worry, restlessness, difficulty concentrating, irritability, muscle tension, and sleep disturbance. A total score of 0–10 is considered minimal, 10–20 as mild, 20–30 as moderate, and >30 as severe [49]. The reliability coefficient of Cronbach’s α of the evaluation tool is 0.96, and the reliability coefficient in this study was 0.90.

#### 2.4.2. Core Life Activities Index

Vitality was evaluated using the ‘Core Life Activities Index (CORE)’. CORE is a tool designed to evaluate the level of vitality in daily activities and includes five questions related to participation in everyday activities, such as sleep, meals, and physical activity. The scores are measured on a 5-point Likert scale (1 = never, 5 = always), where higher scores indicate greater participation in daily activities, interpreted as higher vitality [50]. The reliability coefficient of Cronbach’s α of the evaluation tool is 0.77, and the reliability coefficient in this study was 0.69.

#### 2.4.3. Korean Version of the Satisfaction with the Life Scale

As a tool to evaluate life satisfaction, the Korean version of the Life Satisfaction Scale (K-SWLS) was used. The Life Satisfaction Scale is the Korean version [51] of the satisfaction with life scale developed by Diener et al. [52]. Each item was rated on a 7-point Likert scale (1 = not very much; 7 = very much), and possible life satisfaction scores range from 5 to 35 points. The higher the score, the higher the level of life satisfaction. The reliability coefficient of Cronbach’s α in this study was 0.90.

#### 2.4.4. Perceived Stress Scale

The Korean version of the Perceived Stress Scale (PSS) [53] subjectively evaluates the degree of stress experienced in daily life during the past month. A total of 10 items were used as an evaluation tool that adapted the perceived stress self-perception scale developed by Cohen et al. (1983) [54]. Each item was rated on a 5-point Likert scale (0 = not at all; 4 = very often). The total score ranged from 0 to 40, with higher scores indicating greater perceived stress. The reliability coefficient of Cronbach’s α of the evaluation tool is 0.82, and the reliability coefficient in this study was 0.75.

#### 2.4.5. Parenting Self-Efficacy

The Parenting Efficacy Scale (PSE) is a scale used to evaluate parents’ own beliefs about their ability to perform the parental roles necessary to raise their children in a desirable way. In this study, the PSE developed by Choi and Chung (2001) for elementary school parents in Korea was used [55]. Each item consisted of 37 items that evaluated overall parenting, discipline, communication, healthy parenting, and learning guidance abilities on a 5-point Likert scale (1 = not at all; 5 = very much). The reliability coefficient of Cronbach’s α of the evaluation tool is 0.92, and the reliability coefficient in this study was 0.94.

#### 2.4.6. Children’s Depression Inventory

The Child Depression Inventory (CDI) is a scale for assessing the cognitive, emotional, and behavioral symptoms of childhood depression in children and adolescents aged 8 to 17 years. In this study, the Korean version of the scale, developed by Kovacs (1983) and modified and supplemented by Cho and Lee (1990) to suit Korean children, was used [56]. Each item was rated on a 3-point Likert scale for depressive emotions and behavioral disorders. Total scores range from 0 to 54, with a higher score indicating a higher level of depression. The reliability coefficient of Cronbach’s α of the evaluation tool is 0.88, and the reliability coefficient in this study was 0.71.

### 2.5. Physiological Effect Measures

#### 2.5.1. Scoring Atopic Dermatitis Index

The SCORAD is the most widely used scoring method for evaluating the severity of AD and is used for the standardization of atopic evaluation and interpretation of study results (Figure 6a). The SCORAD index consists of three items: extent of skin lesions (A), severity of lesions (B), and subjective symptoms (C), and the scores of the three items are calculated using the formula A/5 + 7B/2 + C. The total score ranges from 0 to 103, with <15 points indicating mild, 15–40 indicating moderate, and >40 indicating severe [57,58].

#### 2.5.2. Transepidermal Water Loss

TEWL refers to the amount of water lost through the skin. Dry skin due to increased water loss can aggravate pruritus in patients with AD [59]. TEWL is a non-invasive method for evaluating skin barrier function [60]; in this study, TEWL was measured on both forearms using Bio AquaFlux AF200 (Biox Systems, London, UK) (Figure 6b,c).

### 2.6. Statistical Analysis

Statistical analyses were performed using the SPSS 26.0 (IBM Co., Armonk, NY, USA) statistical package. To verify the homogeneity of the caregiver’s experimental group and control group, an independent sample *t*-test was conducted, and normality was verified through the Shapiro–Wilk test; as a result, the parametric statistical method was used. To analyze the reliability of the evaluation tool used in this study, Cronbach’s alpha values were calculated.

Paired *t*-tests (caregivers) and Wilcoxon signed-rank tests (children with AD) were used for pre–post comparisons to verify the effectiveness of the TGP. The effect size for each of the six psychological variables was calculated using Cohen’s d. Cohen’s d is interpreted as a small effect (d = 0.2), medium effect (d = 0.5), or large effect (d = 0.8), following Cohen’s suggestions. Pearson’s correlation coefficient analysis was used to confirm the correlation between variables. At the end of the program, a descriptive statistical analysis was conducted to examine program satisfaction and participants’ perception changes in the experimental group.

## 3. Results

The characteristics of the participants according to their demographics and education levels are shown in Table 1. The average age of the 29 experimental (excluded due to caregiver’s health reasons) and 30 control groups of caregivers of children with AD who participated in the TGP was 42.66 ± 4.25 years old, and more than 80% were highly educated college graduates. As a result of testing the homogeneity of demographic information, the experimental and control groups were homogeneous. The children in the experimental group comprised 15 males (51.7%) and 14 females (48.28%), with an average age of 9.45 ± 3.30 years. An investigation into the medical history of children with AD revealed that over 65% were currently undergoing treatment; the most common drugs used were steroid ointments and lotions. In addition to AD, children had allergic rhinitis (17%), asthma (2%), and food allergies (2%) (Table 2).

### 3.1. Caregivers of Children with AD

After the TGP was conducted for a total of 15 sessions over 3 months, the psychosocial effects on the caregivers of children with AD were measured, and all areas showed positive effects, except for mindfulness perception (Table 3). Depression decreased by 65.52%, from mild to less than mild, and anxiety also decreased by 60.19%, from mild to less than mild. Vitality increased by 14.64%, from below average to above average, and life satisfaction increased by 14.67%. On the stress perception scale, stress decreased by 24.58%, and parenting efficacy increased by 7.14%. The results confirmed the positive effect of TGP participation on the mental health of the caregivers of children with AD, while the control group (NPC), which did not participate in the TGP, did not show significant changes from before to after the program. Even after adjusting for potential confounding factors, such as age, gender, occupation, and education level, to account for potential sources of bias, a significant difference persisted in the experimental group (CCT) (all *p*  <  0.05). Furthermore, in the control group, the effect size (Cohen’s d) was small, whereas the TGP participants exhibited a medium effect size. Specifically, MHS:D (d = 1.08), MHS:A (d = 1.16), and PSS (d = 1.01) showed a large effect size.

As a result of confirming the correlation between mental health variables and parenting efficacy in the caregivers of children with AD (Table 4), the higher the levels of depression, anxiety, and stress, the lower the parenting efficacy. Parenting efficacy had a positive correlation with vitality and satisfaction with life. The lower the depression, anxiety, and stress levels, the higher the parenting efficacy. 

In the subjective surveys (Likert 5-point scale) asking about the effect of participating in the TGP, the participants answered that it was helpful for increased interaction with their children (4.74), increased parenting efficacy (4.69), stress relief (4.65), and improvement in the quality of family life (4.4) (Appendix A).

### 3.2. Children with AD

The impact of TGP participation on children with AD was as follows. The changes in children with AD were assessed before and after their participation in the TGP, without a control group. As a result of measuring the children’s depression scale before and after the TGP, the feeling of depression decreased by 24.44%, from 0.34 ± 0.18 before participating in the TGP to 0.26 ± 0.21 after participating (*p* = 0.033). As a result of analyzing the five subfactors of depression, physiological symptoms, self-abasement, and depressive emotions decreased by more than 30% each, and loss of interest and behavioral disorders decreased by 10%. The SCORAD index, which was conducted to determine the degree of improvement in atopy symptoms, showed that moderate to mild AD symptoms decreased from 18.38 ± 17.84 to 12.76 ± 14.80, an improvement of 30.57% (*p* = 0.025). In the case of TEWL, it decreased by 12.48%, from 24.79 ± 15.71 to 21.70 ± 9.99 after the TGP, but the difference was not significant (*p* = 0.476). No significant correlation was observed between depression and atopic symptoms in children with AD.

## 4. Discussion

This study was conducted to investigate the effect of participation in a TGP on the improvement of psychosocial mental health and AD symptoms of children with AD and their caregivers. Various activities that take place in urban gardens can have a positive impact on children with AD and their caregivers, who are prone to depression and stress. Therefore, by examining various variables together, we have developed a TGP that could help to improve mental health and parenting efficacy. The TGP was planned in stages to enhance interaction and facilitate communication between children and caregivers. Through gardening activities, it was designed to eliminate negative emotions and explore opportunities for change. After participating in a total of 15 sessions of the TGP, as hypothesized, TGP involvement enhanced mental health and improved quality of life [61,62]. Caregivers of children with AD who participated in the TGP reported improvements in six mental health indexes (MHS:D, MHS:A, CORE, K-SWLS, PSE, and PSS) after participation. This supports the results of the evaluation that participation in the TGP had positive influences, such as enhancing interaction with children and parenting efficacy and relieving stress, as evidenced by the results of the satisfaction survey conducted after the TGP was completed. These results are consistent with the therapeutic effects of gardening on mental health reported in previous studies [34,63,64,65,66] 

Additionally, to further examine the relevance of the existing research findings indicating that negative emotions such as depression and stress worsen atopic symptoms, we conducted additional measurements of physiological changes in pediatric AD patients after participating in the TGP. As a result, positive changes were observed in both the SCORAD index and TEWL. In particular, a significant improvement in symptoms was observed in the SCORAD index. Considering the significant decrease in depression in children with AD after TGP participation, and the previous research indicating the impact of mental health on AD symptoms [67,68], it can be speculated that TGP participation contributed to symptom improvement. However, because children were recruited from their primary target, their caregivers, and there was no control group for comparison, it is not possible to conclusively determine the impact of TGP participation on reducing children’s symptoms. Further research is needed for a more conclusive understanding to be formed. Nevertheless, this study suggests that a TGP is effective in enhancing mental health, and this could potentially explain improvements in AD symptoms. Children with AD experience physical symptoms (e.g., itching, eczema, dryness) and associated psychological difficulties (e.g., worry, anxiety, insomnia, depression, stress), which exacerbate AD symptoms in a vicious cycle. Therefore, participating in a TGP, which is designed to promote positive mental health through various interactions with parents while engaging with nature, can be an effective activity for enhancing the psychological well-being of children with AD. This underscores the importance of addressing not only the physical symptoms but also the psychosocial aspects of managing the well-being of children with AD. This study had several limitations. 

First, the sample size of each experimental group was not large, so the possibility of generalization was low, and the impact is thought to be limited in real life. 

To cope with this limitation, we used the G*Power 3.1.9.7 (Heinrich Heine Universität Düsseldorf, Germany) program to calculate the required number of participants (N) for achieving sufficient statistical power and recruited accordingly.

Second, it is possible to predict the positive effects of the decrease in CDI and improvement in AD symptoms after TGP participation. However, the results of the correlation analysis between the two variables were not significant, and this study alone cannot explain a causal relationship with the improvement of AD symptoms. The improvement in AD symptoms in children should be interpreted with caution due to the absence of a comparative control group, and attributing the positive psychological changes solely to the effects of the TGP requires careful consideration. Nevertheless, the possibility that participation in the TGP influenced the symptom improvement still exists. Therefore, it appears essential to conduct a long-term follow-up study focusing on this aspect for further validation.

Third, to accurately assess the impact of participation in Therapeutic Gardening Programs (TGPs) on mental health and the resulting psychophysiological and positive changes in children, it would have been beneficial to conduct comparative studies with a control group. However, the primary subjects of this study were caregivers, and thus there was no control group for comparison with the children. Considering the observed positive changes in this study, future research will explore the inclusion of control groups to address these limitations.

Fourth, in the case of chronic diseases such as AD, it is not easy to observe short-term effects, so long-term observation is required, but, due to the limited research period, the period of this study was short at 3 months. In addition, it should be considered that the symptom improvement and change may not have been significant because the recruited subjects were not severely ill. To overcome these limitations, the development of various TGPs that can provide long-term support for children with AD and their caregivers, and related research, should be continuously conducted. 

Despite these limitations, to the best of our knowledge, our study holds significance as the first piece of clinical research on garden therapy techniques specifically developed for children with atopic dermatitis and their caregivers. Furthermore, as an interdisciplinary study, it demonstrates the potential of the TGP as a psychosocial intervention for those facing challenges due to psychological issues. In recent times, the assessment of quality of life has become increasingly important in the treatment and management of chronic conditions. Therefore, integrated management through psychosocial intervention programs, such as TGPs, which aid in stress management and promote mental well-being, is essential.

## 5. Conclusions

The results of this study showed that the TGP we designed was related to psychosocial well-being and mental health, and, in particular, it was effective at reducing depression in children by reducing stress and increasing parenting efficacy in the caregivers of children suffering from AD. These results suggest that participation in TGPs has a positive psychological and physiological effect on children with AD, as well as the caregivers of children with AD. Therefore, it is necessary to develop and widely implement various TGPs that leverage the advantages of gardens with the aim of promoting psychological stability and mental health. Additionally, future research should focus on understanding the mechanisms of these effects and conduct follow-up evaluations through long-term studies.

## Figures and Tables

**Figure 1 healthcare-12-00919-f001:**
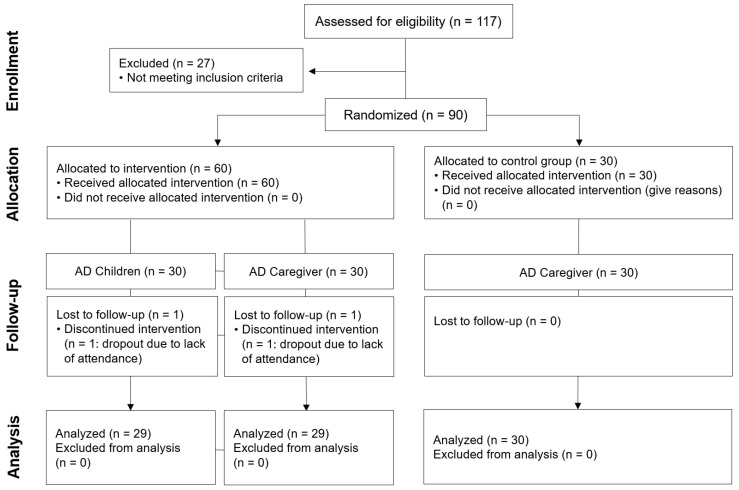
CONSORT flow diagram. Randomized allocation to intervention program (TGP) from enrollment to allocation time and to the follow-up test.

**Figure 2 healthcare-12-00919-f002:**
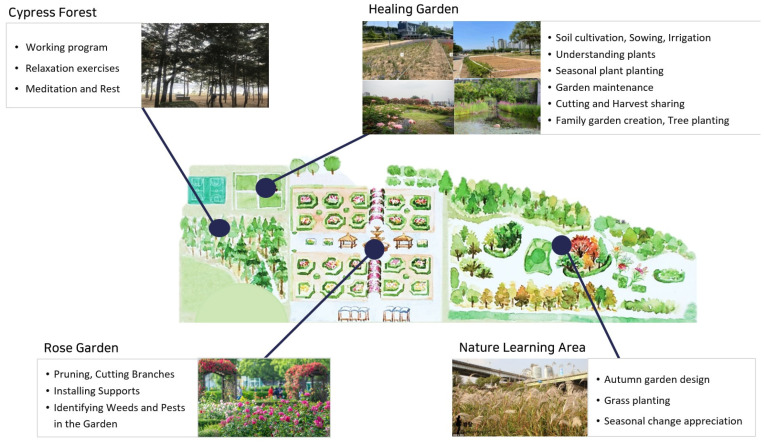
Program composition by activity site.

**Figure 3 healthcare-12-00919-f003:**
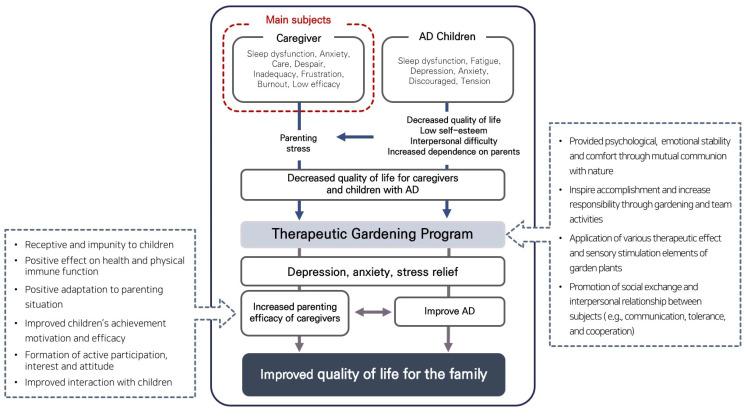
Expected effects of TGP by participant analysis.

**Figure 4 healthcare-12-00919-f004:**
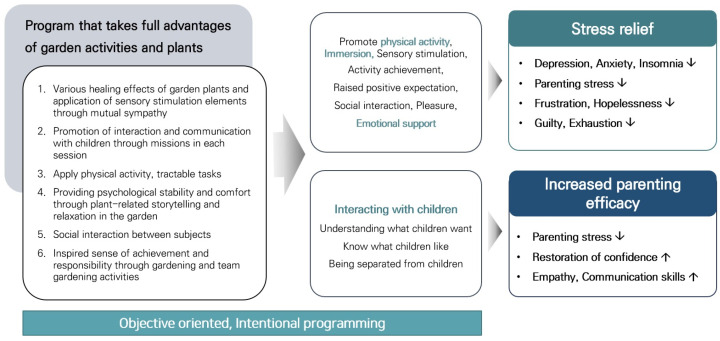
Program composition plan and goals based on target analysis. ↑, increase; ↓, decrease.

**Figure 5 healthcare-12-00919-f005:**
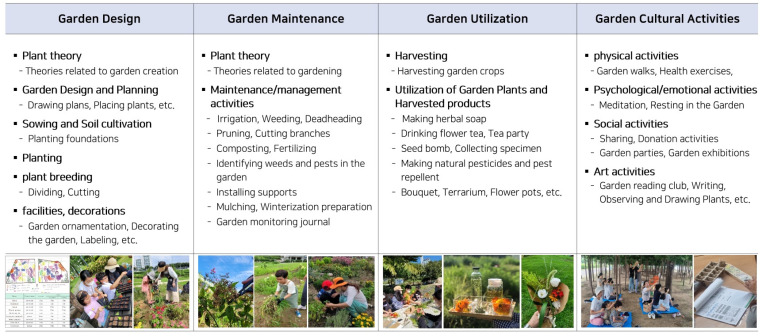
Categorization and activity content for program composition.

**Figure 6 healthcare-12-00919-f006:**
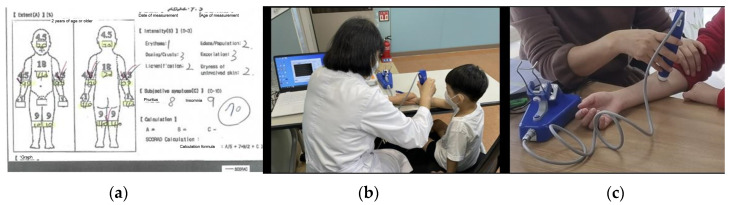
Physiological effect measurement tools (Assessment of Atopic Symptom Improvement): (**a**) Scoring Atopic Dermatitis index (SCORAD index); (**b**,**c**) measurement of transepidermal water loss (TEWL).

**Table 1 healthcare-12-00919-t001:** Clinical characteristics of the participating caregivers (N = 59).

Variables		N (%)	
		CCT	NPC
Age		41.55 ± 4.14	43.73 ± 4.14
Education	High school diploma	3 (10.3)	3 (10.0)
	University degree	23 (79.4)	24 (80.0)
	Graduate school or higher	3 (10.3)	3 (10.0)
Occupation	Housewife	10 (34.5)	9 (30.0)
	Office worker	16 (55.2)	11 (36.7)
	Entrepreneur	2 (6.9)	6 (20.0)
	Other	1 (3.4)	7 (3.3)
Number of Children	1	3 (10.3)	14 (46.7)
	2	17 (58.7)	15 (50.0)
	3 or more	9 (31.0)	1 (3.3)
Age of Children		9.45 ± 3.30	9.9 ± 3.58

Note: CCT, a group of children’s caregivers participating in the therapeutic gardening program; NPC, a group of non-participating caregivers.

**Table 2 healthcare-12-00919-t002:** Disease characteristics of the participating children with AD (N = 29).

Variables	Mean ± SD	N (%)
Age	9.45 ± 3.30	-
Height (cm)	137.9 ± 22.5	-
Weight (kg)	36.6 ± 17.5	-
Current medication use	Steroid ointment or lotion	17 (58.6)
	Antihistamine	3 (10.3)
	Other	2 (6.9)
	No use	7 (24.1)
Comorbid allergic disease	Asthma	2 (6.9)
	Allergic rhinitis	17 (58.6)
	Urticaria	1 (3.4)
	Food allergy	2 (6.9)
	Metal allergy	1 (3.4)
	None	6 (20.7)

Note: SD, standard deviation.

**Table 3 healthcare-12-00919-t003:** Mean scores of the six mental health measures.

Variables	Group	df	Pre-Test	Post-Test	Cohen’s d	t	*p*
			Mean (SD)	Mean (SD)			
MHS:D	NPC	29	9.28 (5.07)	8.46 (7.26)	0.13	0.531	0.531 ^NS^
	CCT	28	8.59 (6.93)	2.96 (2.47)	1.08	4.554	0.000 ***
MHS:A	NPC	29	11.70 (7.34)	11.35 (8.98)	0.42	0.837	0.837 ^NS^
	CCT	28	11.41 (7.33)	4.54 (4.10)	1.16	5.103	0.000 ***
CORE	NPC	29	2.77 (0.66)	2.95(0.58)	0.28	0.214	0.214 ^NS^
	CCT	28	3.04 (0.70)	3.57 (0.76)	0.72	−3.223	0.003 **
K-SWLS	NPC	29	4.15 (1.26)	3.84 (1.25)	0.24	0.148	0.148 ^NS^
	CCT	28	4.09 (1.26)	4.69 (1.10)	0.51	−2.943	0.006 **
PSE	NPC	29	3.73 (0.40)	3.66 (0.44)	0.17	0.205	0.205 ^NS^
	CCT	28	3.71 (0.46)	3.97 (0.48)	0.57	−2.990	0.006 **
PSS	NPC	29	1.90 (0.50)	1.82 (0.51)	0.17	0.373	0.373 ^NS^
	CCT	28	1.83 (0.39)	1.39 (0.49)	1.01	4.093	0.000 ***

Note: ^NS^, **, and *** indicate non-significant or significant at *p* < 0.01, and 0.001. SD, standard deviation; df, degree of freedom; t, t-value; CCT, a group of children’s caregivers participating in the therapeutic gardening program; NPC, a group of non-participating caregivers; MHS:D, Mental Health Screening Tool for Depressive disorders; MHS:A, Mental Health Screening Tool for Anxiety disorders; CORE, Core Life Activities Index; K-SWLS, Korean version of the Satisfaction with Life Scale; PSE, Parenting Self-Efficacy; PSS, Perceived Stress Scale.

**Table 4 healthcare-12-00919-t004:** Result of correlation analysis among seven mental health parameters. Correlations of study variables with parenting self-efficacy (PSE).

Variables	MHS:D	MHS:A	CORE	K-SWLS	PSE	PSS
MHS:D	1					
MHS:A	0.860 **	1				
CORE	−0.543 **	−0.522 **	1			
K-SWLS	−0.425 **	−0.481 **	0.410 **	1		
PSE	−0.480 **	−0.493 **	0.481 **	0.602 **	1	
PSS	0.608 **	0.606 **	−0.624 **	−0.506 **	−0.566 **	1

Note: ** indicate significance at *p* < 0.01. MHS:D, Mental Health Screening Tool for Depressive disorders; MHS:A, Mental Health Screening Tool for Anxiety disorders; CORE, Core Life Activities Index; K-SWLS, Korean version of the Satisfaction with Life Scale; PSE, Parenting Self-Efficacy; PSS, Perceived Stress Scale.

## Data Availability

The datasets generated for this study are available on request to the corresponding author.

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
