# Peer review of "Effect of Therapeutic Gardening Program in Urban Gardens on the Mental Health of Children and Their Caregivers with Atopic Dermatitis"

_healthcare, 2024, doi:10.3390/healthcare12090919_

Round 1
Reviewer 1 Report
Comments and Suggestions for Authors
Thank you for inviting me to complete this review. This is an interesting and valuable paper, with sound research methodology appropriate to the research question. The paper is well written, with appropriate detail in each section. The figures and tables service to enhance the text.
I have just a couple of comments for consideration.
page 6 l184-197 - I think this paragraph could be rephrased and simplified to make more sense. My reading of it is that the sessions were essentially focused on giving parents and children task oriented time together with a mentor to aid communication, but its written in a much more complicated way
did you consider comparing against a standard parenting intervention control group to provide evidence around what gains were due to the TGP and which were simply the result of parents spending time with their children. Please could you consider including this in the limitations section.
Author Response
We sincerely appreciate your constructive and detailed comments. For a clearer explanation as you mentioned, we have made the following modifications to the sentence.
[Line 184-197] Prior to program planning, we categorized and detailed garden activities, as depicted in Figure 5. Our objective was to design various activities within the garden for parents and children, ultimately aiming to transition to a positive psychological state through garden activities.
We incorporated Teli and Gelfand's (1991) parenting efficacy factors, such as helping children understand parents' expectations, recognizing preferences, and facilitating interaction, into garden activities to improve overall parenting efficacy and competence.
During each session, we identified current factors of depression and stress, provided opportunities for interaction, and presented participants and their children with the mission of finding factors that could overcome negative psychological states. These missions included activities using garden plants and herbs, discovering healing plants in the city, and creating a garden that attracts butterflies.
[Line 450-456] To accurately assess the impact of participation in Therapeutic Gardening Programs (TGP) on mental health and the resulting psychophysiological and positive changes in children, it would have been beneficial to conduct comparative studies with a control group. However, the primary subjects of this study were caregivers, and thus there was no control group for comparison with the children. Considering the observed positive changes in this study, future research will explore the inclusion of control groups to address these limitations.

Reviewer 2 Report
Comments and Suggestions for Authors
Although characterized by a certain number of limitations well highlighted by the authors, the research is methodologically correct, the tools are adequately illustrated and the results are clearly stated. The claim that the proposed program of activities appears to have a positive impact on the mental health and well-being of children with AD is entirely plausible.
Strengths: Well organised, well-developed theoretical introduction, clear exposition of the methodology and instruments used, use of appropriate techniques to identify the minimum number of observations, adequate presentation and discussion of results, conclusions congruent with the results obtained, good level of English.
Weakness: that found in the vast majority of field studies i.e. low numbers of observations (although sufficient to draw inferences) and poor generalisability of results. Consequently limited impact on real life.
Author Response
We sincerely appreciate your constructive and detailed comments. As pointed out, while our study provides sufficient evidence for drawing inferences, we acknowledge the limited generalizability to real-life implications. To address these limitations and enhance the practical significance of our findings, we will consider collecting more data and conducting additional analyses. Additionally, we have included explicit mentions of these limitations regarding the limited generalizability in the manuscript to better inform our readers of the constraints of our study.
[Line 436-437] First, the sample size of each experimental group was not large, so the possibility of generalization was low and the impact is thought to be limited in real life.

Reviewer 3 Report
Comments and Suggestions for Authors
I enjoyed this paper very much, I think it will be well received. This type of work is cutting edge and not yet well developed, even in the area of CAM, complementary and alternative medicine. The paper describes a therapeutic gardening program. It is an experimental study of 60 participants, very rare in this area. The majority of the work in this area of environment and positive influences on health is simply case study reports with much fewer quasi-experimental and experimental study designs. That is one reason I enjoyed this paper, including the fact that it was well-written and easy to read.
This is very well written with very few grammatical errors or typos. The typos listed below can be found on lines and should be reviewed/corrected:
Typos/grammatical concerns which should be addressed prior to publication:
In the abstract, "physiological and psychological health of participation" seems awkward. It should likely be "participants" instead of "participation." Abstract page 1
In the sentence "Therefore, it is necessary to develop and widely implement various TGP that leverage the advantages," "TGP" should be "TGP programs" for correct grammar. Page 14
Disregard the three other comments. I had transferred the Adobe file to word and realize now that the line numbers were part of the formatting, not the author's writing.
I also find it disconcerting that there are so many hyphenated words in-general. I did not include that previously as I anticipate it may be an artifact of the style guide or formatting. If it could be reduced, it would be less distracting, but would create an uneven right margin.
The writing is clear, accessible and largely without grammatical error, avoiding highly technical terms (accessible). It is written at a level that encourages broad comprehension making it open to numerous fields, medicine, public health, occupational therapy, psychology, etc. This is yet another benefit of this work and something that is not often found in studies of nature exposure and health outcome.
The materials and methods are detailed and visually/graphically supported and robust. This I find the most satisfying aspect of the writing. By robust I refer to detailed and statistically well planned. Figures 1-6, for example, are very well done and would allow a public health practitioner to easily replicate this study. I am sorry, but I find it difficult to have anything negative to say about this. I do not often see this in the methods of other studies in this area, and again do not often see randomized, controlled intervention studies of this nature.
The statistical analysis is appropriate to the study design and outcome measures. And the verification of data normality (Shapiro-Wilk analysis) and calculation of effect sizes contributes to a robust analysis and study design, also the data collection instrument was evaluated for reliability using a Cronbach's alpha (often a study in and of itself), inspiring confidence in the resulting statistical findings. Both univariate and bivariate fits were used to analyze the correlation between treatment and outcome. The control and treatment groups were also randomly assigned. Only one suggestion is made here, please identify any confounders that were considered as the study design was developed and any measures taken to mitigate them. In other words be clear about inclusion and exclusion criteria.
The findings and discussion are supported by the results reported in that TGP had positive impacts for caregivers and positive feedback from participants, positive impacts on children with AD, a correlation between mental health variables and parenting efficacy, and identified several additional avenues for research as well as identifying the limitations of the current work (sample size, control group of additional children with AD, and caregivers without intervention as these were absent and only 30 non participating caregiver were recruited for control). These issues were acknowledged. Only one additional issue could be addressed, the ability for broad generalizability to countries where significant populations of racial/ethnic minorities reside.
This is an interesting and important study for programming methodology development and studies that seek to quantify contact with natural environments and health outcomes.
Author Response
We sincerely appreciate your constructive and detailed comments. As you mentioned, the grammatical errors pointed out on abstract page 1 and page 14 have been corrected. Confounding factors and measures to alleviate them when selecting experimental and control groups are as follows:
Caregivers who were taking antidepressant medications or undergoing other forms of psychological therapy during the intervention period were excluded, as this could lead to errors in the study. This is mentioned on page 3, lines 103-104 of the paper.
Additionally, we attempted to limit the range of children's ages as much as possible to reduce the confounding factors of various stressful situations experienced by caregivers depending on the time of onset of AD or the child's age.
Caregivers of children who have environmental allergies such as pollen or who are unable to engage in outdoor activities were also excluded, as garden healing involves planting flowers and plants and engaging in outdoor activities. This is mentioned on page 3, line 100-102 of the paper.
